

# River Breezes for Pollutant Dispersion in GoAmazon2014/5

Adan S. S. Medeiros[1,2], Igor O. Ribeiro[1], Marcos V. B. Morais[4], Rita V. Andreoli[3], Jorge A. Martins[5], Leila D. Martins[6], Carla E. Batista[1], Patrícia C. Guimarães[1], Scot T. Martin[*,7], Rodrigo A. F. Souza[*,3].

[1] Post-graduate Program in Climate and Environment, CLIAMB, INPA/UEA, Av. André Araújo, 2936, 69060001, Manaus, Amazonas, Brazil

[2] Amazonas State University, Center of Superior Studies of Tefé, R. Brasília, 2127, 69470-000, Tefé, Amazonas, Brazil

[3] Amazonas State University, Superior School of Technology, Av Darcy Vargas, 1200, 69065020, Manaus, Amazonas, Brazil

[4] Post-graduate Program in Environmental Engineering, Federal University of Technology, Av dos Pioneiros, 3131, 86047-125, Londrina, Paraná, Brazil;

[5] Department of Physics, Federal University of Technology, Av dos Pioneiros, 3131, 86047-125, Londrina, Paraná, Brazil;

[6] Department of Chemistry, Federal University of Technology, Av dos Pioneiros, 3131, 86047-125, Londrina, Paraná, Brazil;

[7] School of Engineering and Applied Sciences, Harvard University, 02138, Cambridge, Massachusets, USA

*E-mail: scot_martin@harvard.edu, souzaraf@gmail.com*

[*]To Whom Correspondence Should be addressed



**Abstract**
The effect of river breezes on pollutant plume dispersion or canalization in the central
Amazon was evaluated. A pollution plume changes atmospheric composition downwind of
Manaus, a city of 2 million people positioned at the confluence between two wide rivers. Herein,
to evaluate the effects of river breezes, two cases were modeled at the mesoscale for March
2014. The first case, "with rivers" (wR), simulated the transport and chemistry of the Manaus
pollution plume as the rivers were in reality. The second case, "without rivers" (woR), carried
out simulations for which all rivers and floodable areas were replaced by forest. The three main
conclusions are as follows: (1) Between the two cases, alterations in wind speeds were maximum
at local noon, and river breezes influenced horizontal wind fields from surface up to 150 m in
altitude, suggesting a capping height of 150 m on most days for the influence of river breezes on
pollutant concentrations. In agreement with this modeling result, data sets collected at 500 m by
aircraft flights showed no apparent influence of the underlying rivers on plume dispersion. The
flights traversed the plume downwind of Manaus during the *Observations and Modeling of the*
*Green Ocean Amazon* (GoAmazon2014/5) Experiment. (2) Between the wR and wOR cases,
changes to downwind concentrations of $O_3$, $NO_x$, and CO pollutants were < 6% as a monthly
average at the supersite "T3" of GoAmazon2014/5, which was 70 km downwind of Manaus and
located between the two main rivers. As single events at T3, maximum one-hour concentration
differences were 39 ppb for $O_3$, 5 ppb for $NO_x$, and 26 ppb for CO. (3) For a focus on the surface
layer of the rivers (0 to 150 m in height), river breezes increased the monthly average $O_3$, $NO_x$,
and CO surface concentrations by 25%, 25%, and <5%, respectively. In addition, strong
canalization occurred 5% of the time based on a difference of 10 ppb in the surface
concentrations of at least two of $O_3$, $NO_x$, and CO between the wR and wOR cases. In
conclusion, although pollutants dispersed by river breezes could at times be a strong effect on
observed pollutant concentrations in the surface river boundary layer, overall most pollution was
transported at heights well above the effects of the river breezes and moved downwind along the
trajectories of the dominant trade winds.



## 1. Introduction

Amazonia represents the single largest hydrographic basin of water volume on Earth
(Sioli, 1984). Land coverage by rivers constitutes 5% of the total 7 million square kilometers of
the Amazon basin during dry season, while in the wet season the rivers increase in horizontal
extent by flooding, reaching a surface coverage of 11% of Amazonia (Hess et al., 2015). An
important confluence of wide rivers occurs nearby Manaus, a city of more than 2 million people
located at {3.0° S, 60.0° W} in the central Amazon (Figure 1). The Rio Negro ("Black River")
flows from the northwest to join the Amazon River, known in Brazil as the Rio Solimões to the
west of Manaus and the Rio Amazonas to the east of Manaus. River width around Manaus varies
from 2 km in narrow sections to 20 km in broader sections.
Wide rivers such as these can induce important atmospheric processes, among which are
river breezes (Oliveira and Fitzjarrald, 1993; Dias et al., 2004; dos Santos et al., 2014). River
breezes arise from the unequal heating of land and water bodies. In the morning, land heats faster
than water, inducing an ascendancy of air over the land and a corresponding subsidence over the
river. In this way, surface winds go from the river toward the land. At an altitude of a few
hundred meters, the circulation cell is closed, and the winds go from the land to the river, with
subsidence over the central portion of the river. The height of the cell depends on the thermal
characteristics of the circulation. At night, the opposite behavior occurs (i.e., the cell reverses)
because the river cools more rapidly than land.
These river breeze circulations at day and night can be important for the local weather
and pollutant dispersion. For instance, during times of weak trade winds Dias et al. (2004) found
that river breeze circulation explained the occurrence of clouds on the eastern bank yet an



absence on the western bank at the confluence between the Amazon and Tapajós rivers, about
600 km to the east of Manaus.

Oliveira and Fitzjarrald (1993, 1994) studied the river breezes in the Manaus region

during the Amazon Boundary Layer Experiments (ABLE) (Garstang et al., 1990; Harriss et al.,
1990). Based on observations of the meridional component of wind speed, the river breezes were
reported as more intense during the dry season than in the wet season, as explained by greater
contrast between river and land temperatures given the greater average insolation of the dry
season. Simulations further suggested that the river breeze induced by the Rio Negro
significantly affected the surrounding daytime surface winds to a distance of 20 km from the
rivers (Oliveira and Fitzjarrald, 1994). The modeled distance was further than initially expected
based on earlier modeling studies, and the key difference appeared to be an improved
representation of the planetary boundary layer (PBL) in the model.

As part of the Large-Scale Biosphere-Atmosphere Experiment in Amazonia-Cooperative

LBA Airborne Regional Experiment-2001 (LBA-CLAIRE-2001), Trebs et al. (2012) traveled by
boat to four locations on the Rio Negro and one on the Solimões River. Daily reversals in surface
winds were attributed to river breezes. Measurements were made of NO, $NO_2$, and $O_3$ surface
concentrations, and pollution was identified at surface river locations from 10 to 150 km
downwind from Manaus. On at least one day, a reversal in wind direction caused by the
afternoon influence of the river breeze was associated with a shift in concentrations
representative of background and polluted conditions. Manaus pollution directed by the river
breezes appeared to be the explanation. The important data sets of this study were, however,
overall sparse (i.e., 8 days of July 2001; Manaus population of 1.2 million at that time), and the
recommendation by the authors was therefore to implement long-term monitoring stations



downwind Manaus and to apply mesoscale modeling to better understand river breeze effects on
the dispersion of the Manaus pollution plume.

In 2014 and 2015, the Observations and Modeling of the Green Ocean Amazon

(GoAmazon2014/5) Experiment was carried out to study the effects of pollutant outflow from
Manaus on atmospheric chemistry, regional climate, and terrestrial ecosystems of an otherwise
typically clean background environment (Martin et al., 2016). Under fair-weather conditions, the
pollution plume was carried westward by equatorial trade winds (Kuhn et al., 2010; Martin et al.,
2017). The GoAmazon2014/5 terrestrial supersite, called "T3", was 70 km to the west of Manaus
(Figure 1).

An important question for the GoAmazon2014/5 experiment was to what extent river

breezes might disperse or canalize Manaus pollution, thereby possibly influencing the
interpretation of data sets collected at the T3 supersite. For a limiting case of full river
canalization, no pollution would reach the T3 site. For an opposite limiting case of weak or no
river breeze effects, all pollution would follow the stable trade winds when fair-weather
conditions prevailed, and air parcels sampled at the T3 site would be interpreted in a fully
Lagrangian framework downwind of the Manaus source region. Between these limiting cases,
partial dispersion of the Manaus pollution plume would be possible. In the context of these
possibilities and in light of the work of Oliveira and Fitzjarrald (1993, 1994) and Trebs et al.
(2012), the study herein presents an analysis of how the rivers around Manaus affect downwind
pollutant dispersion or canalization. The analysis uses mesoscale WRF-Chem modeling and
GoAmazon2014/5 data sets.



## 2. Simulations

*2.1 Model Description*



The *Weather Research and Forecast model coupled with Chemistry* (WRF-Chem) is
described by Grell et al. (2005). Version 3.6.1 was used for the present study. A two-domain
configuration was used (Medeiros et al., 2017). The inner domain represented an area of 302 km
× 232 km, had a horizontal resolution of 2 km × 2 km, and had 38 vertical layers from ground to
160 hPa. The outside boundaries of the inner domain were forced by data from an outer domain.
The outer domain represented an area of 1050 km × 800 km, had a resolution of 10 km × 10 km,
and had 38 vertical layers from ground to 160 hPa. Both domains were centered on {2.908° S,
60.319° W}. The meteorology of the outside boundary of the outer domain was forced by the
*Climate Forecast System Reanalysis* (CFSv2) product of the National Center for Environment
Prediction (NCEP) at a temporal resolution of 6 h and a spatial resolution of 0.5° (Saha et al.,
2011). The inputs of surface temperature were also considered based on CFSv2 product. The
chemical composition of the outside boundary of the outer domain was forced by the Model *for*
*Ozone and Related chemical Tracers* (MOZART-4) (Emmons et al., 2010).
Data of the Moderate Resolution Imaging Spectroradiometer (MODIS) satellite at a
resolution of 500 m were used for land cover (Channan et al., 2014). These data were used as
obtained for the case of "with rivers" (wR). For the case of "without rivers" (woR), the rivers
and main flooded areas of MODIS land cover were replaced by forest in the pre-processor of the
WRF-Chem model.
The physics parametrizations used in the simulations were described previously (Ying et
al., 2009; Misenis and Zhang, 2010; Gupta and Mohan, 2015), including for the study region in
the central Amazon (Medeiros et al., 2017). The parametrizations treated the physics of the



surface layer (Grell et al., 1994), the land surface (Chen et al., 1997), the boundary layer (Hong
et al., 2006), shortwave radiation (Chou and Suarez, 1999), longwave radiation (Mlawer et al.,
1997), cloud microphysics (Lin et al., 1983), and cumulus clouds (Grell and Freitas, 2014). At
Figure S1, the comparison between observed and simulated temperature, relative humidity and
wind speed at "T3" supersite show that the simulations performed herein represent the diurnal
cycle of these variables.
For chemical parametrizations, the *Model of Emissions of Gases and Aerosols from*
*Nature* (MEGAN, version 2.1) (Guenther et al., 2012) was used for biogenic emissions.
Anthropogenic emissions from transport, power, and industry for Manaus in 2014 were based on
the emission inventory of Medeiros et al. (2017). The *Regional Acid Deposition Model*
(RADM2) was used to simulate gas-phase chemistry (Chang et al., 1989).
2.2 *Model Runs*
Simulations of the wR and woR cases were carried out for all days in March 2014. Other
characteristics between the two simulations remained the same. This approach aimed to isolate
the river breeze effects on the transport of pollutants downwind of Manaus. For time zero, the
inner and outer domains were both initialized to CFSv2 and MOZART-4. A spin-up time of 24 h
was used, which was sufficient to fully replace the air of the inner domain. After the spin-up
period, simulations in lots of 72 h were performed for March 2014 as a balance between
conserving computing resources and avoiding excessive numerical drift (Medeiros et al., 2017).
**3. Data Sets**
Data sets were collected during the first intensive operating period (IOP1) of the
GoAmazon2014/5 project by instrumentation of the G-159 Gulfstream I (G-1) of the ARM
Aerial Facility (AAF) of the USA Department of Energy (Schmid et al., 2014; Martin et al.,



2017). Concentrations of $O_3$ (Thermo Scientific Model 49i), $NO_x$ (airborne $NO_x$ analyzer, Air
Quality Devices), and CO (Los Gatos 23r) were measured. The aircraft performed 16 flights
during IOP1. Data sets of two flights (March 14 and 21) were chosen for analysis herein based
on flight tracks over both river and land while cutting across the Manaus pollution plume at an
altitude of approximately 500 m. There were no flights at lower altitude.
**4. Results and Discussion**
4.1 *Height of River Breeze Circulation Cell*
The effect of river breezes on horizontal wind speeds was evaluated. As monthly means,
the left column of Figure 2 presents the wR case, and the right column shows the differences
between the wR and woR cases. The rows represent plots at surface, 100-m altitude, and 500-m
altitude. Comparison between columns shows that the river breezes significantly affected mean
surface wind speeds but that the effects decreased with altitude.
The change of horizontal wind speeds with altitude is presented in detail in Figure 3
through height cross sections along points A, B, and C above the Rio Negro nearby Manaus (cf.
Figure 1). Panels in Figure 3 show wind speed differences between the wR and woR cases for all
times as well as for 00:00, 06:00, 12:00, and 18:00 (local time). In the absence of solar radiation
(i.e., at 00:00 and 18:00 local), the differences in horizontal wind speeds were relatively small.
The strongest differences were at noon corresponding to maximum daily solar irradiance, as
expected, because of the largest thermal gradients between land and river at these times (Oliveira
and Fitzjarrald, 1993, 1994; Dias et al., 2004; Fitzjarrald et al., 2008; de Souza and dos Santos
Alvalá, 2014; dos Santos et al., 2014; de Souza et al., 2016). The river breeze effect was
confined to less than 150 m, as shown in the monthly average plot of Figure 3. Across individual
days, the maximum and minimum heights for significant noontime river breeze effects were 300





and 60 m, respectively (results not shown). Overall, the results of Figures 2 and 3 lead to the
conclusion that the river breeze effect on wind speeds was confined on most days to below 150
m in altitude, even under high noontime solar irradiance.

For comparison, aircraft data sets of $O_3$, $NO_x$, and CO concentrations from 500-m altitude

are plotted in Figure 4 (Martin et al., 2017). Carbon monoxide was mostly inert on the time
scales of the simulations, oxides of nitrogen were significantly lost during downwind transport,
and ozone was a secondary pollutant rapidly produced within Manaus and over the nearby rivers,
quickly reaching steady-state concentrations (Medeiros et al., 2017; Rafee et al., 2017). The
panels of the left column of Figure 4 show that the flight paths intercepted the Manaus pollution
plume in the planetary boundary layer on March 14 from 10:20 to 11:20 (local time; UTC - 4 h).
The panels of the right column show that interception took place on March 21 from 13:00 to
14:00. Below each map, concentrations along the flight tracks are plotted, and the red shading
represents times that the aircraft was over a river. The results show that there was no obvious
influence of river breezes on the dispersion of the Manaus pollutant plume at 500 m. Although
Figure 4 shows only two flights, which were selected for substantial data coverage over both
river and land during a single flight, all 16 flights from March 2014 were investigated, and a
strong river breeze effect was not apparent in any of them (analysis not shown). The aircraft data
sets thus corroborate the tendencies represented in Figures 2 and 3 that the effects of rivers
presence on plume dispersion are confined to the surface boundary layer over the rivers,
typically below 150 m.
4.2 *Effects of River Breezes on Downwind Concentrations*

The "T3" GoAmazon2014/5 terrestrial supersite was located at {-3.2133 N, -60.5987 E},

approximately 70 km to the west of Manaus and in the dominant direction of prevailing trade





winds that passed through Manaus (Figure 1). The panels in the top row of Figures 5, 6, and 7
show $O_3$, $NO_x$, and CO concentrations, respectively, at the T3 location for the wR case as the
blue line, the woR case as the red line, and their difference as the black line. The pollutant
concentrations did not change greatly in the presence or absence of the rivers. Quantitatively, the
perturbations caused by the presence of the rivers on the $O_3$, $NO_x$, and CO concentrations were
less than 6%. Maximum one-hour concentration differences were 39 ppb for $O_3$, 5 ppb for $NO_x$,
and 26 ppb for CO across the month. The overall implication is that the effects of the trade winds
on transport largely dominated over the influence of river breezes in this region when
considering the larger part of Manaus pollutant outflow, in agreement with the modeling and
observational results of section 4.1.
4.3 *Surface River Concentrations*

Two locations "R1" and "R2" were chosen to evaluate pollutant dispersion and

canalization in the river surface layer (0 to 150 m). Based on the prevailing trade winds location
R1 at {-3.0699 N, -60.2199 E} consistently intercepted the urban outflow (Figure 1). By
comparison, location R2 at {-2.9800 N, -60.4901 E} was nominally outside of the trajectories of
the trade winds that passed over and then downwind of Manaus. For the analysis, the simulated
pollutant concentrations at R2 were compared to those at R1 in the wR and woR cases to test the
extent of river canalization of the plume, meaning transport of air parcels along the river rather
than in the prevailing direction of the synoptic-scale trade winds.

The time series of the simulated $O_3$, $NO_x$, and CO concentrations at the R1 and R2

locations are plotted in Figures 5, 6, and 7. Blue lines show the wR case, red lines show the woR
case, and black lines represent their difference. At R1, the differences of (wR - woR) were
significant, in particular for concentrations of $O_3$ (+28.4%) and $NO_x$ (+26.0%) (Table 1, monthly





means). Strong differences were also simulated at R2 for $O_3$ (+25.5%) and $NO_x$ (+25.6%). The
river breezes thus shifted the surface level dispersion pattern of the Manaus plume. At both sites,
the differences were smaller (< 5%) for CO concentrations because regional background
concentrations exceeded the Manaus contribution. Absolute changes (± ppb) for four time
periods across the day are listed in Table S1 of the Supplementary Material. The difference
between the wR and woR cases exceeded 10 ppb at R2, which can be called strong canalization,
for at least two pollutants at 5% frequency for March 2014.

An example of strong canalization occurred for the simulation on March 2 at 13:00 (local

time). The three columns of Figure 8 show the wR case, the woR case, and their difference, all at
surface altitude. The three rows show results for $O_3$, $NO_x$, and CO. For the wR case, a small
portion of the pollution plume traversed the course of Rio Negro to the northwest, following the
local wind field. A boat collecting surface samples would report increases of $O_3$ by 20 ppb and of
CO by 10 ppb. Overall, comparison of the wR and woR case shows that the simulated plume was
tighter with less transverse spreading in the absence of rivers. In both the wR and woR cases,
most pollution followed the predominant direction of the trade winds, and the effects of river
canalization on diverting the plume were small by comparison.

In summary, this study evaluated the effects of river breezes on pollutant plume

dispersion or canalization in the central Amazon. The simulations showed that the horizontal
monthly mean wind speeds were significantly altered by river breezes to altitudes of 75 m as
monthly 24-h means and to 150 m for monthly averages across the maximum noontime solar
radiation. The strongest effects of river breezes on pollutant dispersion were thus from 0 to 150
m above the river surface. Aircraft data confirmed the absence of observable effects of river
breezes on plume dispersion at flight level (500 m). Overall, the perturbation on pollutant



concentrations downwind of Manaus at "T3" GoAmazon2014/5 supersite was modeled as below
6%, consistent with pollutant transport mostly above 150 m. At river surface level, the monthly
average $O_3$, $NO_x$, and CO surface concentrations increased by 25%, 25%, and <5%, respectively.
Differences in surface river concentrations exceeded 10 ppb for at least two pollutants at a
frequency of 5% for March 2014. The overall conclusion is that the Manaus pollution plume
dispersion could at times be partially canalized leading to significant changes of surface river
concentrations even as most pollution passed overhead of the river circulation cell and
dominantly followed trajectories along the prevailing overhead trade winds.

**Acknowledgments**

We acknowledge support from the Central Office of the Large-Scale Biosphere-Atmosphere

Experiment in Amazonia (LBA), the National Institute of Amazonian Research (INPA), the

Amazonas State University (UEA), the Brazilian Innovation Agency (FINEP), and the

Atmospheric System Research (ASR) program of the Office of Biological and Environmental

Research, Office of Science, United States Department of Energy (DOE) (grant DE-

SC0011115). A. Medeiros thanks the Brazilian Federal Agency for Support and Evaluation of

Graduate Education (CAPES) for the grant scholarship, linked to the doctoral program in

Climate and Environment (CLIAMB). Funding was received under grant 062.00568/2014 of the

Amazonas State Research Foundation (FAPEAM). A FAPEAM grant of a Senior International

Visiting Researcher is also acknowledged. The work was conducted under scientific licenses

001030/2012-4 and 400063/2014-0 of the Brazilian National Council for Scientific and

Technological Development (CNPq).

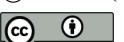


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





**List of Figures**

**Figure 1.** Satellite image of the Manaus region. The Rio Negro comes from the northwest, and the Rio Solimões arrives from the west. The confluence of the two rivers is to the southeast of Manaus, beginning the Amazon River ("Rio Amazonas"). Yellow markers show locations of (i) the measurement supersite called "T3", (ii) two river locations "R1" and "R2" considered in the modeling methodology herein, and (iii) points "A", "B", and "C" along a river cross section, also used in the methodology herein.

**Figure 2.** Mean horizontal wind speeds for March 2014. The left column represents the base case wR ("with rivers"). The right column represents the difference case (wR - woR) ("with rivers compared to without rivers"). Rows represent altitude as (1) near surface, (2) 100 m, and (3) 500 m.

**Figure 3.** Difference in mean horizontal wind speed for (wR - woR) ("with rivers compared to without rivers"). Plots are shown as vertical cross sections along points A, B, and C of the Figure 1 as follows: (a) for all of March 2014, (b) at 00:00, (c) at 06:00, (d) at 12:00, and (e) at 18:00, all in local time (UTC - 4 h).

**Figure 4.** Concentrations of $O_3$, $NO_x$, and CO measured by instrumentation on board an aircraft during GoAmazon2014/5 at an altitude of approximately 500 m (Martin et al., 2017). Concentrations are plotted in false color, and the legends on the right-hand side of each row show the scaling. Below each main panel, a line plot shows the concentrations marked by points A through H along the flight paths. Red shading demarcates periods when the aircraft was flying over a river.

**Figure 5.** Time series of $O_3$ concentrations at the T3, R1, and R2 locations. The left column plots



the cases of wR ("with rivers"; blue) and woR ("without rivers"; red). The right

column shows the difference in concentrations as (wR - woR).

**Figure 6.** Time series of NO$_x$ concentrations at the T3, R1, and R2 locations. The left column

plots the cases of wR ("with rivers"; blue) and woR ("without rivers"; red). The right

column shows the difference in concentrations as (wR - woR).

**Figure 7.** Time series of CO concentrations at the T3, R1, and R2 locations. The left column

plots the cases of wR ("with rivers"; blue) and woR ("without rivers"; red). The right

column shows the difference in concentrations as (wR - woR).

**Figure 8.** Near-surface concentrations of (a) O$_3$, (b) NO$_x$, and (c) CO (March 2, 13:00 local time,

UTC - 4 h). The first and second columns represent the cases of wR ("with rivers")

and woR ("without rivers"), respectively. The vector field in each panel shows the

near-surface horizontal winds. The third column shows the difference in

concentrations as (wR - woR). For reference, the locations of T3, R1, and R2 are

marked.



**Table 1.** Percent change in pollutant concentration $X$ for the case of wR ("with rivers") compared to that of woR ("without rivers"), calculated as $(X_{wR} - X_{woR})/X_{wR}$, where $X$ is the monthly mean at a location T3, R1, or R2 for each of $O_3$, $NO_x$, and CO.

|     | $O_3$   | $NO_x$  | CO     |
| --- | ------- | ------- | ------ |
| T3  | +5.5%   | -4.6%   | +1.2%  |
| R1  | +28.4%  | +26.0%  | +2.9%  |
| R2  | +25.5%  | +25.6%  | +2.6%  |



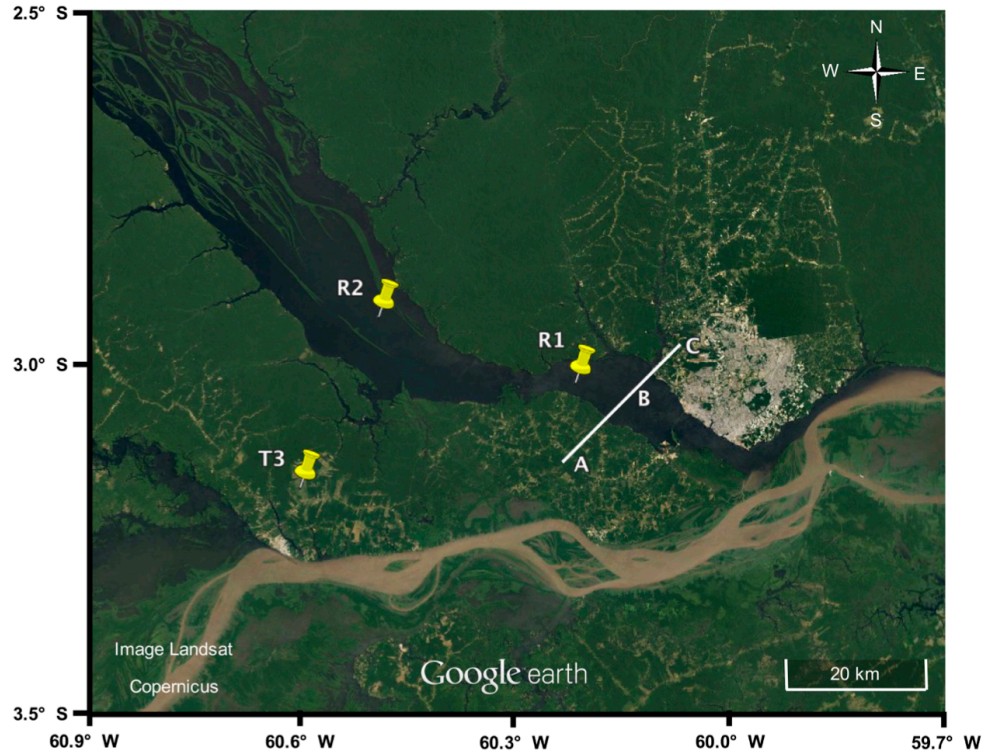

Figure 1



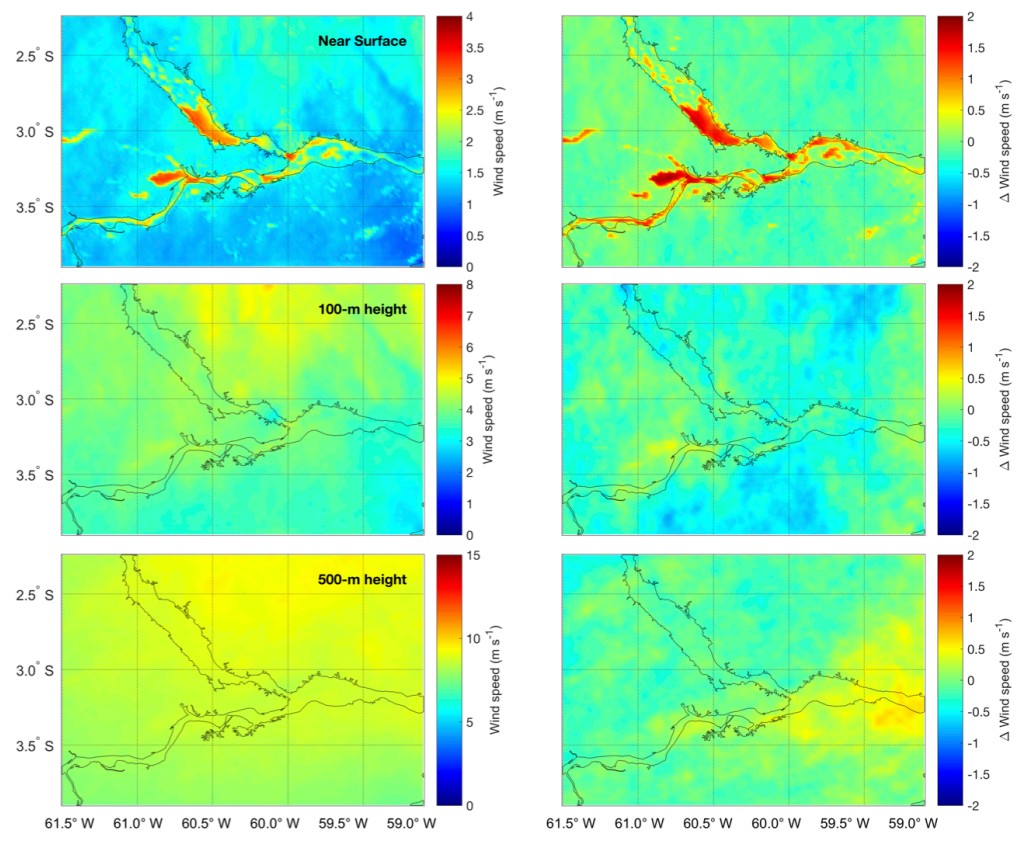

Figure 2





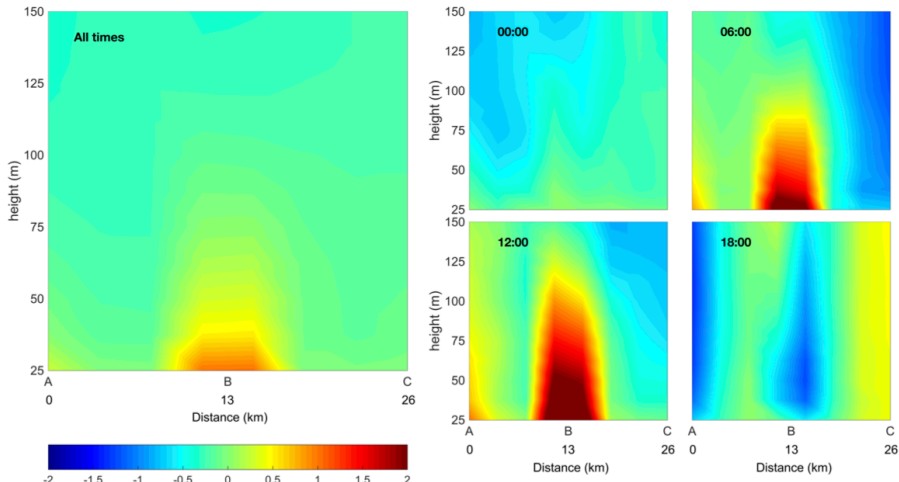

Figure 3



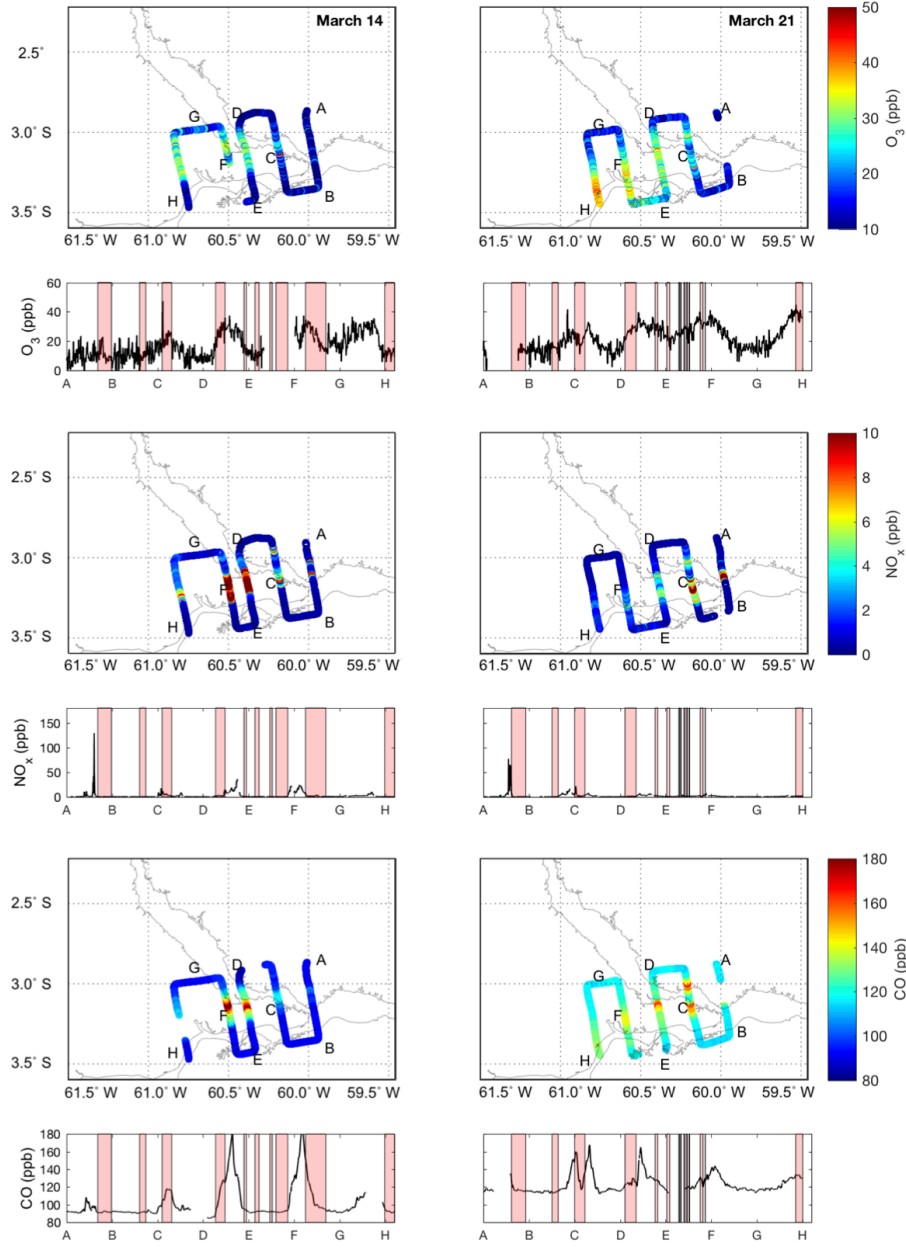

Figure 4



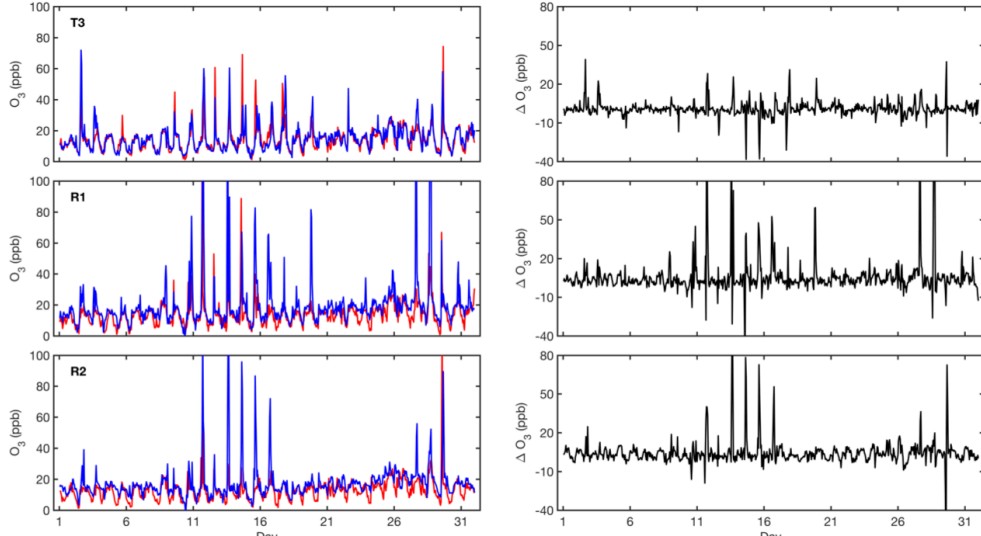

Figure 5





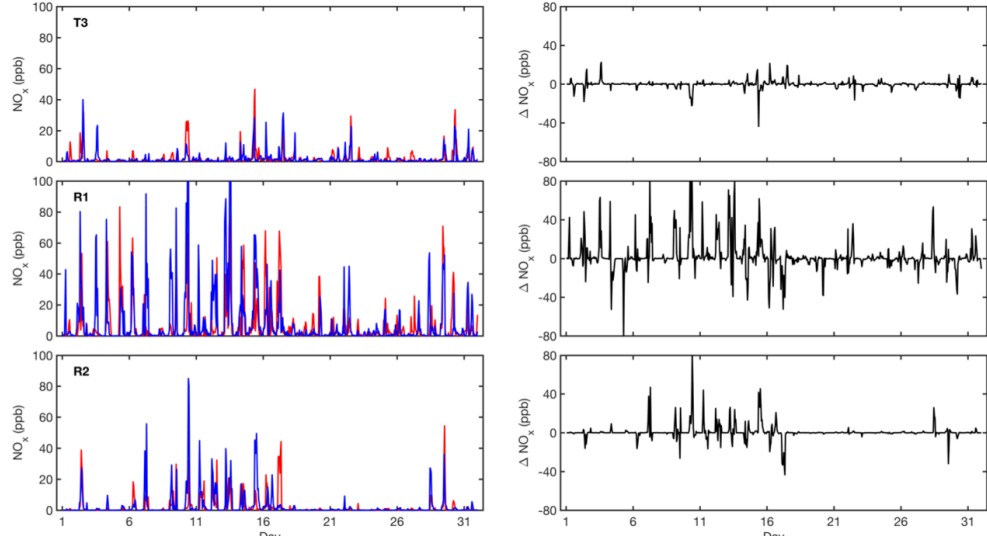

Figure 6



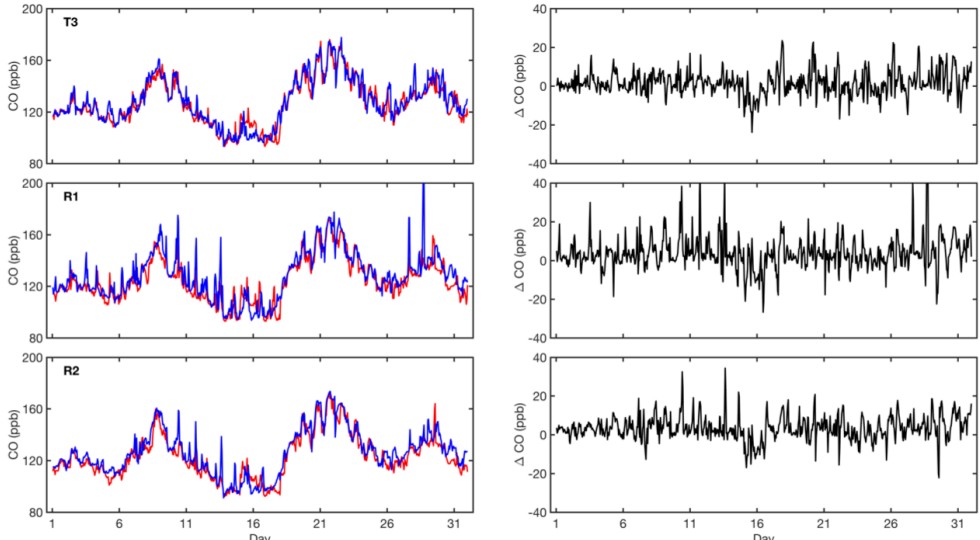

Figure 7







Figure 8