# Peer review of "River Breezes for Pollutant Dispersion in GoAmazon2014/5"

_Atmospheric Chemistry and Physics, 2018_

## Referee Comment (RC1) · Anonymous Referee #1 · 4 Jun 2018

General Comment

This study analyzed the effect of the river breezes on the dispersion or canalization of the Manaus pollution using observational data and numerical simulations with high spatial resolution. Results show that the river breeze cell is, on average, confined below 150 m suggesting that the river breeze effect on pollution dispersion above this level is negligible. Observed data of CO NOx and O3 concentration confirmed that the river breeze does not affect the pollution dissipation at 500 m. On the other hand, the river breezes remarkably affect the pollutant dispersion near the surface, mainly over "R1" and "R2" locations. This study demonstrated that the river breezes play an important role in the Manaus pollution dispersion in the low atmospheric levels. Moreover, this paper also highlights locations where the river breeze influence on pollution dispersion

is more effective. These finds are very important to the Amazon local-climate understanding and complement the previous discoveries. The used methods are appropriate for the study purpose and the paper, in general, is well written. However, some "major" points should be reviewed. The points are below listed.

Specific Comments

Introduction

1 - In the paragraph that starts in line #38, I recommend citing the dos Santos, M. J., M. A. F. Silva Dias, and E. D. Freitas (2014),doi:10.1002/2014JD021969, since this paper shows evidence of river breeze for the Manaus area using long-term observation.

2 – The citation of dos Santos, M. J., M. A. F. Silva Dias and E. D. Freitas (2014) is also recommended in the paragraph that starts in the line #52;

3 – This part of the text should be improved. "On at least one day, a reversal in wind direction caused by the afternoon influence of the river breeze was associated with a shift in concentrations representative of background and polluted conditions". It is not clear.

Simulations

Some crucial information required in local circulations modeling are missing in this section:

**1 - There is no information about the soil initialization. In other words, the soil initial conditions (i.e., soil temperature and moisture) used in these simulations have to be described in this section.**

**2 - There is no information about the surface temperature of the rivers prescribed in the simulations.**

In line 125 It is written that the simulations were carried out for March 2014, but there It does not explain why this period was selected.

The following sentence is not clear at all. "After the spin-up period, simulations in lots of 72 h were performed for March 2014 as a balance between conserving computing resources and avoiding excessive numerical drift (Medeiros et al., 2017)."

Results and Discussion

In line 166 It is written: "Figure 4 shows that the flight paths intercepted the Manaus pollution plume in the planetary boundary layer on March 14 from 10:20 to 11:20 (local time; UTC - 4 h)". Figure 4 show does not show It since there is no time information there.

In lines 186 and 187, I suggest you present the maximum absolute concentration differences instead of maximum concentration differences. If you present the maximum absolute concentration, the NOx difference value will be larger than 5 ppb.

In lines 170-171 It is written: "The results show that there was no obvious influence of river breezes on the dispersion of the Manaus pollutant plume at 500 m". This affirmation can be corroborated by plotting vertical cross-sections of the simulated pollutants across Manaus pollutant plume. Thus, I suggest to include these cross-sections on the paper discussion.

In lines 187-190 I agree with the following sentence: "The overall implication is that the effects of the trade winds on transport largely dominated over the influence of river breezes in this region when considering the larger part of Manaus pollutant outflow". But, I do not agree with this part: "in agreement with the modeling and observational results of section 4.1." In section 4.1, It showed the concentration of pollutants at 500 m altitude, where the river breeze effect is negligible. In section 4.2, It was analyzed the pollutant concentration near the surface where the local circulations are remarkable. In other words, the conditions are different.

The small perturbations (concentration difference less than 6%, line 186 ) caused by the presence of the rivers is, probably, related to the river breeze activity. The river

breeze occurrence is more frequent in the dry season, please check, dos Santos M. J., M. A. F. Silva Dias, and E. D. Freitas (2014).

In lines 209-210 the following sentence is unclear. "which can be called strong canalization, for at least two pollutants at 5% frequency for March 2014."

In Figure 2 and 8, I suggest that you replot the right column figures using a Diverging Color Schemes for a better visualization of the results. The following link shows many options. https://www.mathworks.com/matlabcentral/fileexchange/34087-cbrewer—colorbrewer-schemes-for-matlab

In the captions of Figure 5,6 and 7, you should mention the atmospheric level of the presented pollutant concentrations.
* * *

---

## Referee Comment (RC2) · Anonymous Referee #2 · 9 Jun 2018

The manuscript shows the effect of river breezes for the city of Manaus using data from a field campaign and WRF-Chem modeling. I believe this is the first study to analyze the effects of river breezes on pollutant transport at the city scale using models, which makes it very valuable. The paper is well written and the topic is relevant and in the scope of the Journal. I have a few comments and suggestions outlined below, mainly requesting additional complementary analysis.

**General comments**

Given that this is the first study of it's kind, it would be great to see additional analysis performed by including sensitivity simulations. Given that river breezes are being studied, it would be helpful to see what's the model response to the representation of river breezes when the grid-spacing is changed. I recommend adding simulations

(with and without rivers) going down to 1 km horizontal resolution and by refining the vertical resolution in the region where the breeze develops. Including the 10km domain into this analysis (if two-way nesting is not used) would be helpful too. Analyzing the changes in meteorology and pollutant concentrations as a function of the grid-spacing would provide great insight into the subject.

Another sensitivity simulation that comes to mind has to do with changes of the river properties. Fig. 1 shows a very distinct albedo difference between the Rio Negro vs the Amazons. Adding a simulation where this contrast is added by changing the reflective properties of one of the rivers could be useful to see if this has any effect.

Finally, the manuscript should be revised more carefully for English by a native English speaker.

Comments by line

19. You could include the % of change associated with these concentrations

Intro. Additional information of the effect of lake breezes on air quality could be included together with expected similarities and differences to river breezes.

99. Provide the vertical grid-spacing in the first layers up to 200m, as this is the region where the largest sensitivities occur due to river breezes

Fig S1. It would be helpful to add the data for both simulations in this plot, to have an idea of the changes (if any) in the performance of modeled meteorology due to river breeze

133-135. You mean that every 72h meteorology was initialized from CFSv2? If this is the case, did you leave some spin-up time for meteorology after every initialization? This would be recommended given that you are studying river breeze effects.

Fig. 2. Maybe you can add a column showing the mean surface values at noon, which is where your maximum changes occur.

**ACPD**
147-165. This analysis is great but could be complemented with the addition of changes of wind direction and vertical winds. Maybe adding arrows representing horizontal (in the A-C axis) and vertical winds (differences?) in the profiles could help. In the intro you mention that the breezes happen due to temperature gradients, so including temperature differences would also contribute.

166-182. Here you only discuss the flight observations. I recommend adding the simulated values over the flight track for both simulations on the time series in Figure 4, so is visually clear that there are no differences at this altitude and also to have an idea of the model performance.

Figs 5-7. It would be great to have an idea of model performance by adding the observed T3 concentrations to the upper-left panel

184-195. CO seems to present a high frequency variability coming from Manaus while there is a long frequency probably because of background air changing, could you elaborate on this? Is this also seen in the observations at T3?

**ACPD**

---

## Referee Comment (RC3) · D. R. Fitzjarrald (Referee) · 13 Jun 2018

Reviewed by: David R. Fitzjarrald, Atmospheric Sciences Research Center, University at Albany, SUNY

General comments.

The topic addressed by this paper is so important to the overall project that it deserves to be done more thoroughly. I always fear that, in the course of large field projects, topics are doled out for publishing among participating groups in such a way to avoid overlapping discussions, but, in so doing, incomplete research results. Fortunately, the authors have considerable information in hand, and, if they are willing to re-enter the data analysis phase a bit, they could complete the missing parts. The resulting paper

will be of greater use to the community. Please make a major revision. Among the missing parts I identified are:

• Plots and discussion of the vertical structure of the convective boundary layer (CBL), and this material must include wind direction & speed as well as potential temperature. There are three ways to address this issue with the available data: The conventional radiosondes at MAO, the aircraft soundings on takeoff and upon landing, and the model output. The authors make a conclusion that breeze circulations are not important in dispersion, on the two case study days, but they offer little context. How thick was the convective boundary layer, over land and over the river, where vertical mixing will be suppressed?

• Better exploitation of the single month of modeling than was done. The current side-long glance at these results is insufficient. I am looking particularly for average diurnal behavior near the surface, and all I get is Figure S1, showing wind speed, temperature and RH. Since an RH plot looks like the upside-down rendering of the temperature plot over the diurnal cycle, one would do better showing the specific humidity. Better still, plot the temperature and specific humidity hourly medians on the same plot, reserving the right vertical axis for specific humidity and saturation specific humidity. (It's not that difficult, though the latter has a slightly nonlinear scale, of course.) More important is some indication of the diurnal variation of wind direction, so that the reader can understand how the breeze baroclinicity affects directional wind shear.

• A focus on the river breeze overshadows the other effect of the rivers, mainly that the flow is channeled. (Kindly write 'channeling' rather than the word 'canalization', which, though a word in English and happily a cognate to its Portuguese counterpart, is almost never used.) How might the dispersion of the pollution plume change as it goes over the river, where surface fluxes must diminish as compared to over the land? Make some estimates of the buoyancy flux over the river and compare with the direct measurements over the land.

Specific comments.

1. Abstract, line 3. Did I miss something? Where do the authors discuss how the chemical composition of the plume changes downstream?

2. Abstract, lines 9-13. If the authors' model result indicates that any breeze circulations might be confined below the 150-m level, was the design of the project, with flights limited to 500 m altitude, flawed? Also, was any effort done to consider lighter-wind vs. strong-wind conditions. One might think the breeze circulations would be more apparent in the former case. Please comment.

3. Abstract lines 25-27. The authors conclude that "..most pollution was transported at heights well above the effects of the river breezes..." What then does this indicate about the wisdom of locating the vast DOE ARM resources at a single site at the surface, downwind? Please comment on this and elaborate about what these sentences mean. How is the pollution 'information' communicated to the surface? Answering this question brings you straight back to trying to understand the differences between the CBL over land and whatever is present over the rivers.

4. Line 48. Prof. Maria A. F. Silva Dias, has been cited for some time as "Silva Dias". Please follow this pattern, so that readers looking up cited papers will not get confused.

5. Line 166. On what time scale can one conclude that "carbon monoxide was mostly inert on the time scale of the simulations..."? Here are some competing scales: a) Mixing in the convective boundary layer $z_i/w^*$, where $w^*$, the convective velocity scale that depends on the surface buoyancy flux. This might be different over the river & over the land; b) the time it takes the air to advect from Manaus to the point of measurement; and of course, c) time of day (time since the surface layer became convective, though one might argue that over the river that layer was convective all night.
* * *

---

## Author Comment (AC1) · 31 Jul 2018

Response to Review #1 1 - Comment from Referee: This study analyzed the effect of the river breezes on the dispersion or canalization of the Manaus pollution using observational data and numerical simulations with high spatial resolution. Results show that the river breeze cell is, on average, confined below 150 m suggesting that the river breeze effect on pollution dispersion above this level is negligible. Observed data of CO NOx and O3 concentration confirmed that the river breeze does not affect the pollution dissipation at 500 m. On the other hand, the river breezes remarkably affect the pollutant dispersion near the surface, mainly over "R1" and "R2" locations. This study demonstrated that the river breezes play an important role in the Manaus pollution dispersion in the low atmospheric levels. Moreover, this paper also highlights locations

where the river breeze influence on pollution dispersion is more effective. These finds are very important to the Amazon local-climate understanding and complement the previous discoveries. The used methods are appropriate for the study purpose and the paper, in general, is well written. However, some "major" points should be reviewed. The points are below listed.

1 - Author's response: The authors thank the reviewer for the careful reading of the manuscript and the valuable input that was provided.

2 - Comment from Referee: In the paragraph that starts in line #38, I recommend citing the dos Santos, M. J., M. A. F. Silva Dias, and E. D. Freitas (2014),doi:10.1002/2014JD021969, since this paper shows evidence of river breeze for the Manaus area using long-term observation.

2 - Author's response: The citation is added to the revised manuscript.

3 - Comment from Referee: The citation of dos Santos, M. J., M. A. F. Silva Dias and E. D. Freitas (2014) is also recommended in the paragraph that starts in the line #52;

3 - Author's response: The citation is added in the suggested paragraph, followed by the sentence:

3 – Author's changes in Manuscript: Line 54: "Oliveira and Fitzjarrald (1993, 1994) studied the river breezes in the Manaus region during the Amazon Boundary Layer Experiments (ABLE) (Garstang et al., 1990;Harriss et al., 1990). Based on observations of the meridional component of wind speed, the river breezes were reported as more intense during the dry season than in the wet season, as explained by greater contrast between river and land temperatures given the greater average insolation of the dry season. Simulations further suggested that the river breeze induced by the Rio Negro significantly affected the surrounding daytime surface winds to a distance of 20 km from the rivers (Oliveira and Fitzjarrald, 1994). The modeled distance was further than initially expected based on earlier modeling studies, and the key difference appeared

to be an improved representation of the planetary boundary layer (PBL) in the model. More recently, dos Santos et al. (2014) studied the river breeze near Manaus, concluding that this local circulation affects wind patterns and, consequently, spatial/temporal distribution of the precipitation on the region."

4 - Comment from Referee: This part of the text should be improved. "On at least one day, a reversal in wind direction caused by the afternoon influence of the river breeze was associated with a shift in concentrations representative of background and polluted conditions". It is not clear.

4 - Author's response: The revised text is clarified as follows:

4 - Author's changes in Manuscript: Line 72: "On at least one day, a reversal in wind direction caused by the afternoon influence of the river breeze was associated with an increase in pollutant concentrations."

5 - Comment from Referee: There is no information about the soil initialization. In other words, the soil initial conditions (i.e., soil temperature and moisture) used in these simulations have to be described in this section.

5 - Author's response: We thank the reviewer for this important comment. The soil initial conditions were obtained from Climate Forecast System Reanalysis (CFSv2) product of the National Center for EnvironmentÂăPrediction (NCEP). The following clarification is added to the manuscript:

5 - Author's changes in Manuscript: Line 109 "The meteorology of the outside boundary of the outer domain was forced by the Climate Forecast System Reanalysis (CFSv2) product of the National Center for Environment Prediction (NCEP) at a temporal resolution of 6 h and a spatial resolution of 0.5° (Saha et al., 2011). The inputs of surface temperature, and soil conditions were also considered based on CFSv2 product."

6 - Comment from Referee: The following sentence is not clear at all. "After the spin-up period, simulations in lots of 72 h were performed for March 2014 as a balance between

conserving computing resources and avoiding excessive numerical drift (Medeiros et al., 2017)."

6 - Author's response: We thank the reviewer for this observation. The intention of the sentence was to explain that the simulations were performed in groups of 72 h. This approach provided a balance between computational time and numerical diffusion. The revised text is clarified as follows:

6 - Author's changes in Manuscript: Line 136: "Simulations of the wR and woR cases were carried out for all days in March 2014. Other characteristics between the two simulations remained the same. This approach aimed to isolate the river breeze effects on the transport of pollutants downwind of Manaus. For time zero, the inner and outer domains were both initialized to CFSv2 and MOZART-4. The simulations were performed in groups of 96 h, with 24 h of spin-up followed by 72 h of valid run, as described in Medeiros et al. (2017)"

7 - Comment from Referee: In line 166 It is written: "Figure 4 shows that the flight paths intercepted the Manaus pollution plume in the planetary boundary layer on March 14 from 10:20 to 11:20 (local time; UTC - 4 h)". Figure 4 show does not show It since there is no time information there.

7 - Author's response: We thank the reviewer for the observation. The points A and H were considered as begin and end of the paths for March 14 (i.e. left panels) and March 21 (i.e. right panels), and the time that the measurements were performed corresponds to 10:20 for point A and 11:20 for point H. The caption of Figure 4 is revised, as follows:

7 - Author's changes in Manuscript: "Caption of Figure 4: Concentrations of O3, NOx, and CO, for (left) March 14 and (right) March 21 from 10:20 and 11:20 (local time, UTC – 4 h), measured by instrumentation on board an aircraft during GoAmazon2014/5 at an altitude of approximately 500 m (Martin et al., 2017). Concentrations are plotted in false color, and the legends on the right-hand side of each row show the scaling. Below each main panel, a line plot shows the concentrations marked by points A (10:20)

through H (11:20) along the flight paths. Red shading demarcates periods when the aircraft was flying over a river."

8 - Comment from Referee: In lines 186 and 187, I suggest you present the maximum absolute concentration differences instead of maximum concentration differences. If you present the maximum absolute concentration, the NOx difference value will be larger than 5 ppb.

8 - Author's response: We thank the reviewer for this suggestion. In order to prevent any misinterpretation from general readers, the mentioned sentence is removed from the manuscript.

9 - Comment from Referee: In lines 170-171 It is written: "The results show that there was no obvious influence of river breezes on the dispersion of the Manaus pollutant plume at 500 m". This affirmation can be corroborated by plotting vertical cross-sections of the simulated pollutants across Manaus pollutant plume. Thus, I suggest to include these cross-sections on the paper discussion.

9 - Author's response: We thank the reviewer for this suggestion. The Figure "S3" is inserted in the supplement, and the manuscript is revised, as follows:

9 - Author's changes in Manuscript: Line 183: "The results show that there was no obviousÂăinfluence of river breezes on the dispersion of the Manaus pollutant plume at 500 m, which is corroborated by cross sections of O3, NOx, and CO concentrations shown in Figure S3."

10 - Comment from Referee: In lines 187-190 I agree with the following sentence: "The overall implication is that the effects of the trade winds on transport largely dominated over the influence of river breezes in this region when considering the larger part of Manaus pollutant outflow". But, I do not agree with this part: "in agreement with the modeling and observational results of section 4.1." In section 4.1, It showed the concentration of pollutants at 500 m altitude, where the river breeze effect is negligible.

In section 4.2, It was analyzed the pollutant concentration near the surface where the local circulations are remarkable. In other words, the conditions are different.

10 - Author's response: We thank the reviewer for this perspective. The mentioned agreement is related to the following line of thinking. The aircraft data do not appear to support a river breeze effect on Manaus pollution plume at 500 m. This observation can be explained by the conclusions in section 4.1 that changes in horizontal winds are confined to near the surface in the first 150 m of altitude. The presence of the rivers does not interfere on pollutant dispersion at 500 m. This result is corroborated by Figure "S3", which shows the cross-section of the considered pollutants.

11 - Comment from Referee: The small perturbations (concentration difference less than 6%, line 186 ) caused by the presence of the rivers is, probably, related to the river breeze activity. The river breeze occurrence is more frequent in the dry season, please check, dos Santos M. J., M. A. F. Silva Dias, and E. D. Freitas (2014).

11 - Author's response: The authors agree with the reviewer. In order to enrich the discussion, the following sentence is inserted in the revised manuscript:

11 - Author's changes in Manuscript: Line 243: "Differences in surface river concentrations exceeded 10 ppb for at least two pollutants at a frequency of 5% for March 2014. In caveat, this value represents the wet season and might differ for the dry season (dos Santos et al., 2014)."

12 - Comment from Referee: In lines 209-210 the following sentence is unclear. "which can be called strong canalization, for at least two pollutants at 5% frequency for March 2014."

12 - Author's response: For clarification, the following sentence is inserted in the manuscript:

12 - Author's changes in Manuscript: Line 221: "The difference between the wR and woR cases exceeded 10 ppb at R2 for at least two pollutant at 5% frequency for March

2014. These conditions were considered as a strong channeling cases."

13 - Comment from Referee: In Figure 2 and 8, I suggest that you replot the right column figures using a Diverging Color Schemes for a better visualization of the results. The following link shows many options. https://www.mathworks.com/matlabcentral/fileexchange/34087-cbrewer—colorbrewer-schemes-for-matlab

13 - Author's response: We thank the reviewer for this valuable input. In this regard, please see improved Figures 2 and 8.

14 - Comment from Referee: In the captions of Figure 5,6 and 7, you should mention the atmospheric level of the presented pollutant concentrations.

14 - Author's response: Information concerning height of the pollutant concentrations is added to the captions of Figures 5, 6, and 7, as follows:

14 - Author's changes in Manuscript: Caption of Figure 5: "Time series of O3 near-surface concentrations at the T3, R1, and R2 locations. The left column plots the cases of wR ("with rivers"; blue) and woR ("without rivers"; red). The right column shows the difference in concentrations as (wR - woR)." Caption of Figure 6: "Time series of NOx near-surface concentrations at the T3, R1, and R2 locations. The left column plots the cases of wR ("with rivers"; blue) and woR ("without rivers"; red). The right column shows the difference in concentrations as (wR - woR)." Caption of Figure 7: "Time series of CO near-surface concentrations at the T3, R1, and R2 locations. The left column plots the cases of wR ("with rivers"; blue) and woR ("without rivers"; red). The right column shows the difference in concentrations as (wR - woR)."

References

dos Santos, M. J., Silva Dias, M. A., and Freitas, E. D.: Influence of local circulations on wind, moisture, and precipitation close to Manaus City, Amazon Region, Brazil, Journal of Geophysical Research: Atmospheres, 119, 13,233-213,249, 2014.

Garstang, M., Ulanski, S., Greco, S., Scala, J., Swap, R., Fitzjarrald, D., Martin, D., Browell, E., Shipman, M., and Connors, V.: The Amazon boundary-layer experiment (ABLE 2B): A meteorological perspective, Bulletin of the American Meteorological Society, 71, 19-32, 1990.

Harriss, R., Garstang, M., Wofsy, S., Beck, S., Bendura, R., Coelho, J., Drewry, J., Hoell, J., Matson, P., and McNeal, R. J.: The Amazon boundary layer experiment: wet season 1987, Journal of Geophysical Research: Atmospheres, 95, 16721-16736, 1990.

Martin, S. T., Artaxo, P., Machado, L., Manzi, A. O., Souza, R. A. F., Schumacher, C., Wang, J., Biscaro, T., Brito, J., Calheiros, A., Jardine, K., Medeiros, A., Portela, B., Sá, S. S. d., Adachi, K., Aiken, A. C., Albrecht, R., Alexander, L., Andreae, M. O., Barbosa, H. M. J., Buseck, P., Chand, D., Comstock, J. M., Day, D. A., Dubey, M., Fan, J., Fast, J., Fisch, G., Fortner, E., Giangrande, S., Gilles, M., Goldstein, A. H., Guenther, A., Hubbe, J., Jensen, M., Jimenez, J. L., Keutsch, F. N., Kim, S., Kuang, C., Laskin, A., McKinney, K., Mei, F., Miller, M., Nascimento, R., Pauliquevis, T., Pekour, M., Peres, J., Petäjä, T., Pöhlker, C., Pöschl, U., Rizzo, L., Schmid, B., Shilling, J. E., Dias, M. A. S., Smith, J. N., Tomlinson, J. M., Tóta, J., and Wendisch, M.: The Green Ocean Amazon Experiment (GoAmazon2014/5) Observes Pollution Affecting Gases, Aerosols, Clouds, and Rainfall over the Rain Forest, Bull. Am. Meteorol. Soc., 98, 981-997, 10.1175/bams-d-15-00221.1, 2017.

Medeiros, A. S. S., Calderaro, G., Guimarães, P. C., Magalhaes, M. R., Morais, M. V. B., Rafee, S. A. A., Ribeiro, I. O., Andreoli, R. V., Martins, J. A., Martins, L. D., Martin, S. T., and Souza, R. A. F.: Power plant fuel switching and air quality in a tropical, forested environment, Atmos. Chem. Phys., 17, 8987-8998, 10.5194/acp-17-8987-2017, 2017.

Oliveira, A. P., and Fitzjarrald, D. R.: The Amazon river breeze and the local boundary layer: I. Observations, Boundary-Layer Meteorology, 63, 141-162, 1993.

Oliveira, A. P., and Fitzjarrald, D. R.: The Amazon river breeze and the local boundary

layer: II. Linear analysis and modelling, Boundary-Layer Meteorology, 67, 75-96, 1994.

Saha, S., Moorthi, S., Wu, X., Wang, J., Nadiga, S., Tripp, P., Behringer, D., Hou, Y.-T., Chuang, H.-y., Iredell, M., Ek, M., Meng, J., Yang, R., Mendez, M. P., van den Dool, H., Zhang, Q., Wang, W., Chen, M., and Becker, E.: NCEP Climate Forecast System Version 2 (CFSv2) 6-hourly Products, in, Research Data Archive at the National Center for Atmospheric Research, Computational and Information Systems Laboratory, Boulder, CO, 2011.

---

## Author Comment (AC2) · 31 Jul 2018

Response to Review #2 General Comment: The manuscript shows the effect of river breezes for the city of Manaus using data from a field campaign and WRF-Chem modeling. I believe this is the first study to analyze the effects of river breezes on pollutant transport at the city scale using models, which makes it very valuable. The paper is well written and the topic is relevant and in the scope of the Journal. I have a few comments and suggestions outlined below, mainly requesting additional complementary analysis.

1 – Comment from Referee: Given that this is the first study of its kind, it would be great to see additional analysis performed by including sensitivity simulations. Given that river breezes are being studied, it would be helpful to see what's the model response to the representation of river breezes when the grid-spacing is changed. I recommend adding simulations (with and without rivers) going down to 1 km horizontal resolution and by refining the vertical resolution in the region where the breeze develops. Including the 10km domain into this analysis (if two-way nesting is not used) would be helpful too. Analyzing the changes in meteorology and pollutant concentrations as a function of the grid-spacing would provide great insight into the subject.

1 – Author's response: We thank the reviewer for this perspective. We agree that sensitivity simulations to grid spacing can be interesting and helpful to understand atmospheric processes. In the study design, we adjusted the grid to 2 km in the central region of the simulation in an around the rivers. Internally (i.e., without mention in the manuscript), we also carried out simulations at 3 km and 5 km while scoping the project. The rivers have a width of 5 to 10 km, so the model at 2 km resolution captures the major features of the river breezes, specifically as related to penetration of the river breeze to higher altitudes (in which 19 vertical levels were used in the first 500 m) and possible pollutant dispersion of the nearby urban area (scale of 20 km). Other scientific questions could require different scales. The revised text includes this caveat, as follows:

1 - Author's changes in Manuscript: Line 103: "The inner domain represented an area of 302 km x 232 km, had a horiziontal resolution of 2 km x 2 km, and had 38 vertical layers from ground to 160 hPa. The first 14 levels were below 200 m in altitude. With this grid configuration, the major features of the river breeze could be represented."

2 - Comment from Referee: Another sensitivity simulation that comes to mind has to do with changes of the river properties. Fig. 1 shows a very distinct albedo difference between the Rio Negro vs the Amazons. Adding a simulation where this contrast is added by changing the reflective properties of one of the rivers could be useful to see if this has any effect.

2 - Author's response: We thank the reviewer for this perspective. This suggestion is

very interesting in a scientific perspective geared toward understanding weak versus strong river breeze effects. The focus of our study, however, was to understand the effects of river breezes as they are now around Manaus on possible regional pollutant dispersion and/or channeling. The study of reflective properties of the rivers would be a good effort for a future manuscript focused more on the physics of river breezes than the effects of the presence of the rivers on pollution plume dispersion, as herein.

3 - Comment from Referee: Finally, the manuscript should be revised more carefully for English by a native English speaker.

3 - Author's response: The revised manuscript has been carefully reviewed by a native English speaker

4 - Comment from Referee: Line 19. You could include the % of change associated with these concentrations

4 – Author's response: We thank the reviewer for this perspective. After consideration, however, we think that the values shown in line 19 are associated with single events and the proposed insertion of percentage values can confuse the reader. In addition, Figures 5, 6, and 7 show the time series of pollutant concentrations at T3, R1, and R2, so the reader can have an idea of how much these individual events are significant at mean basis for March 2014.

5 - Comment from Referee: Intro. Additional information of the effect of lake breezes on air quality could be included together with expected similarities and differences to river breezes;

5 - Author's response: The following sentence is added to the manuscript.

5 – Author's changes in the Manuscript: Line 39: "River ăbreezes arise from the un-equal heating of land and water bodies. In the morning, land heats faster ăthan water, inducing an ascendancy of air over the land and a corresponding subsidence over the ăriver. In this way, surface winds go from the river toward the land. At an altitude

of a few ă hundred meters, the circulation cell is closed, and the winds go from the land to the river, with subsidence over the central portion of the river. The height of the cell depends on the thermal characteristics of the circulation. At night, the opposite behavir occurs (i.e., the cell reverses) because the river cools more rapidly than land. At the size scale of the regional rivers, these processes are similar to lake breezes that also arise from the thermal gradient between land and water, especially in the Amazon region where there are expansive wetlands during the wet season (Walter, 1973;Hess et al., 2015;Moura et al., 2004)."

6 - Comment from Referee: Line 99. Provide the vertical grid-spacing in the first layers up to 200m, as this is the region where the largest sensitivities occur due to river breezes;

6 - Author's response: The authors agree with the reviewer. To address the point, the following sentence is added to the manuscript. Author's changes in Manuscript: Line 103: "The inner domain represented an area of 302 km x 232 km, had a horiziontal resolution of 2 km x 2 km, and had 38 vertical layers from ground to 160 hPa. The first 14 levels were below 200 m in altitude."

7 - Comment from Referee: Fig S1. It would be helpful to add the data for both simulations in this plot, to have an idea of the changes (if any) in the performance of modeled meteorology due to river breeze;

7 - Author's response: The authors thank the reviewer for this suggestion. We think that these kind of analysis is out of the scope of the article because we intend to focus on the effects of the presence of the rivers on pollution plume dispersion.

8 - Comment from Referee: Line 133-135. You mean that every 72h meteorology was initialized from CFSv2? If this is the case, did you leave some spin-up time for meteorology after every initialization? This would be recommended given that you are studying river breeze effects.

8 - Author's response: We thank the reviewer for this observation. The simulations were performed in groups of 96 h, with 24 h of spin-up, and using 72 h of valid run. The text is clarified as follows:

8 - Author's changes in Manuscript: Line 136: "Simulations of the wR and woR cases were carried out for all days in March 2014. Other characteristics between the two simulations remained the same. This approach aimed to isolate the river breeze effects on the transport of pollutants downwind of Manaus. For time zero, the inner and outer domains were both initialized to CFSv2 and MOZART-4. The simulations were performed in groups of 96 h, with 24 h of spin-up followed by 72 h of valid run, as described in Medeiros et al. (2017)"

9 - Comment from Referee: Fig. 2. Maybe you can add a column showing the mean surface values at noon, which is where your maximum changes occur.

9 - Author's response: The authors thank the reviewer for the suggestion. We think that the variability of the horizontal wind speed is shown and discussed in Figure 3.

10 - Comment from Referee: Line 147-165. This analysis is great but could be complemented with the addition of changes of wind direction and vertical winds. Maybe adding arrows representing horizontal (in the A-C axis) and vertical winds (differences?) in the profiles could help. In the intro you mention that the breezes happen due to temperature gradients, so including temperature differences would also contribute.

10 - Author's response: We thank the reviewer for this valuable input. We did the analysis of the changes in vertical winds and wind direction while scoping this project, but conclusive results were not possible from monthly means. For the temperature gradients, a Figure S2 is added to the manuscript. The manuscript is clarified as follows:

10 - Author's changes in Manuscript: Line 163: "The strongest differences were at noon corresponding to maximum daily solar irradiance, as expected, because of the largest

thermal gradients between land and river at these times (Oliveira and Fitzjarrald, 1993, 1994;Silva Dias et al., 2004;Fitzjarrald et al., 2008;dos Santos et al., 2014;de Souza and dos Santos Alvalá, 2014;de Souza et al., 2016). Monthly mean surface temperature are shown in Figure S2 for wR and woR cases. The thermal gradient is higher at noon rather than midnight."

11 - Comment from Referee: Line 166-182. Here you only discuss the flight observations. I recommend adding the simulated values over the flight track for both simulations on the time series in Figure 4, so is visually clear that there are no differences at this altitude and also to have an idea of the model performance.

11 - Author's response: The authors thank the reviewer for this recommendation. However, we think that a full evaluation of the model performance is out of scope of this manuscript. The focus of the manuscript is to describe the effects of the presence of the rivers on Manaus plume dispersion. A different manuscript with this kind of evaluation is the topic of some current work and might be forthcoming later. The model configuration used in the simulations is based in Medeiros et al. (2017), which compares the observed and simulated ozone concentrations for Manaus region.

12 - Comment from Referee: Figs 5-7. It would be great to have an idea of model performance by adding the observed T3 concentrations to the upper-left panel

12 - Author's response: We thank the reviewer for this suggestion. We think that this comparison is highly valuable, yet it is out of the scope of the manuscript. In this regard, please see the answer for point 11.

13 - Comment from Referee: Line 184-195. CO seems to present a high frequency variability coming from Manaus while there is a long frequency probably because of background air changing, could you elaborate on this? Is this also seen in the observations at T3?

13 - Author's response: The reviewer is correct that there is low-frequency variability

because of changing background air. The same variability is seen upwind at the ATTO tower. This variability must be addressed in the context of a global model because of CO outflow from biomass burning in Africa as well as CO production from VOC photo-oxidation (i.e., isoprene). For purposes of our regional study, the background CO concentration is a boundary condition, i.e., CO has a relatively long atmospheric lifetime.

References

de Souza, D. O., and dos Santos Alvalá, R. C.: Observational evidence of the urban heat island of Manaus City, Brazil, Meteorol. Applic., 21, 186-193, 10.1002/met.1340, 2014.

de Souza, D. O., dos Santos Alvalá, R. C., and do Nascimento, M. G.: Urbanization effects on the microclimate of Manaus: A modeling study, Atmos. Res., 167, 237-248, 10.1016/j.atmosres.2015.08.016, 2016.

dos Santos, M. J., Silva Dias, M. A., and Freitas, E. D.: Influence of local circulations on wind, moisture, and precipitation close to Manaus City, Amazon Region, Brazil, J. Geophys. Res, 119, 13,233-213,249, 10.1002/2014jd021969, 2014.

Fitzjarrald, D. R., Sakai, R. K., Moraes, O. L., Cosme de Oliveira, R., Acevedo, O. C., Czikowsky, M. J., and Beldini, T.: Spatial and temporal rainfall variability near the Amazonâ ĂŘTapajós confluence, J. Geophys. Res., 113, 10.1029/2007JG000596, 2008.

Hess, L. L., Melack, J. M., Affonso, A. G., Barbosa, C., Gastil-Buhl, M., and Novo, E. M.: Wetlands of the lowland Amazon basin: Extent, vegetative cover, and dual-season inundated area as mapped with JERS-1 synthetic aperture radar, Wetlands, 35, 745-756, 10.1007/s13157-015-0666-y, 2015.

Medeiros, A. S. S., Calderaro, G., Guimarães, P. C., Magalhaes, M. R., Morais, M. V. B., Rafee, S. A. A., Ribeiro, I. O., Andreoli, R. V., Martins, J. A., Martins, L. D., Martin, S. T., and Souza, R. A. F.: Power plant fuel switching and air quality in a tropical, forested
environment, Atmos. Chem. Phys., 17, 8987-8998, 10.5194/acp-17-8987-2017, 2017.

Moura, M. A. L., Meixner, F. X., Trebs, I., Lyra, R. F. d. F., Andreae, M. O., and Nascimento Filho, M. F. d.: Observational evidence of lake breezes at Balbina lake (Amazonas, Brazil) and their effect on ozone concentrations, Acta Amazonica, 34, 605-611, 10.1590/S0044-59672004000400012, 2004.

Oliveira, A. P., and Fitzjarrald, D. R.: The Amazon river breeze and the local boundary layer: I. Observations, Bound.-Layer Meteorol., 63, 141-162, 10.1007/BF00705380, 1993.

Oliveira, A. P., and Fitzjarrald, D. R.: The Amazon river breeze and the local boundary layer: II. Linear analysis and modelling, Bound.-Layer Meteorol., 67, 75-96, 10.1007/BF00705508, 1994.

Silva Dias, M., Dias, P. S., Longo, M., Fitzjarrald, D. R., and Denning, A. S.: River breeze circulation in eastern Amazonia: observations and modelling results, Theor. Appl. Climatol., 78, 111-121, 10.1007/s00704-004-0047-6, 2004.

Walter, A.: Detailed Mesometeorological Studies of Air Pollution Dispersion in the Chicago Lake Breeze, Mon. Weather Rev., 101, 387, 10.1175/1520-0493(1973)101<0387:DMSOAP>2.3.CO;2, 1973.

---

## Author Comment (AC3) · 31 Jul 2018

Response to Review #3

1 - Comment from Referee: The topic addressed by this paper is so important to the overall project that it deserves to be done more thoroughly. I always fear that, in the course of large field projects, topics are doled out for publishing among participating groups in such a way to avoid overlapping discussions, but, in so doing, incomplete research results. Fortunately, the authors have considerable information in hand, and, if they are willing to re-enter the data analysis phase a bit, they could complete the missing parts. The resulting paper will be of greater use to the community. Please make a major revision. Among the missing parts I identified are:

[Figure]

1 - Author's response: The authors thank the reviewer for the acknowledgement of the importance of the manuscript.

2 - Comment from Referee: Plots and discussion of the vertical structure of the convective boundary layer (CBL), and this material must include wind direction & speed as well as potential temperature. There are three ways to address this issue with the available data: The conventional radiosondes at MAO, the aircraft soundings on takeoff and upon landing, and the model output. The authors make a conclusion that breeze circulations are not important in dispersion, on the two case study days, but they offer little context. How thick was the convective boundary layer, over land and over the river, where vertical mixing will be suppressed?

2 - Author's response: The reviewer's goal for the manuscript and the authors' goal for the manuscript appear to differ.

As authors, we already have the observational result: A result of GoAmazon2014/5, which was unexpected given extrapolation of results of previous modeling studies in the region, was the absence of an effect of the rivers in aircraft measurements of gaseous and particle tracers of pollution at 500 m before, over, and after the river (i.e., in direction of typical easterlies of the trade winds).

Therefore, the purpose of the study is to assess if current modeling capabilities and understand capture the absence of a river breeze effect on the dispersion of the Manaus pollutant outflow.

In this regard, the previous modeling studies that led to the above-mentioned extrapolation were not precisely focused on the same question as ours, they used different grid sizes, and they employed older physics and parameterizations. The intention of the present study was therefore to provide a modeling study focused on the river to test if our current understanding of physics of this environment and approach to its modeling was consistent with these aircraft observations. The conclusion of the manuscript is that the physics embedded in the model suggests at limit of 150 m in the height of

the river effects.

The reviewer's interests for model-measurement closure appear broader than the purpose of the present manuscript, and we also appreciate the intellectual interest in these broader questions. Nevertheless, these broader interests appear outside the need of the focus and scope of conclusions of the present manuscript.

The abstract and introduction are revised to clarify the scope of the manuscript and to avoid the confusion reflected in this comment/reply between reviewer/authors for the general readership.

3 - Comment from Referee: Better exploitation of the single month of modeling than was done. The current side-long glance at these results is insufficient. I am looking particularly for average diurnal behavior near the surface, and all I get is Figure S1, showing wind speed, temperature and RH. Since an RH plot looks like the upside-down rendering of the temperature plot over the diurnal cycle, one would do better showing the specific humidity. Better still, plot the temperature and specific humidity hourly medians on the same plot, reserving the right vertical axis for specific humidity and saturation specific humidity. (It's not that difficult, though the latter has a slightly nonlinear scale, of course.) More important is some indication of the diurnal variation of wind direction, so that the reader can understand how the breeze baroclinicity affects directional wind shear.

3 - Author's response: As with comment 2, we appreciate the reviewer's broader interest, as reflected in, "I am looking particularly for," but the authors' interest for this manuscript is different. Please see reply 2. As mentioned there, the revised abstract and introduction for the readership clarifies the intention of the authors' study. We do agree with the reviewer that additional important and interesting studies and analyses of river breezes can be made, even as we judge that the provided story is sufficient to support that state-of-the-art modeling approaches support the observation of the absence of a river breeze effect on Manaus pollutant dispersion on most days at most

times as a broad statement, with exceptions at certain days and at certain times, as discussed in the manuscript.

4 - Comment from Referee: A focus on the river breeze overshadows the other effect of the rivers, mainly that the flow is channeled. (Kindly write 'channeling' rather than the word 'canalization', which, though a word in English and happily a cognate to its Portuguese counterpart, is almost never used.)

4 - Author's response: "Canal" is adjusted to "channel" throughout the manuscript.

5 - Comment from Referee: How might the dispersion of the pollution plume change as it goes over the river, where surface fluxes must diminish as compared to over the land? Make some estimates of the buoyancy flux over the river and compare with the direct measurements over the land.

5 - Author's response: The purpose of the manuscript is to respond to this question: "How might the dispersion of the pollution plume change as it goes over the river?" The conclusions state: "This study evaluated the effects of river breezes on pollutant plume dispersion or channeling in the central Amazon." The results plotted in Figure 3 show that the effective buoyancy modeled for the river "peters out" at 150 m. The G-1 aircraft flights directly over the river at 500 m (which was the lowest allowed legal height for flights at that time) show no evidence of an underlying river in the gas and particle tracers of the pollution.

6 - Comment from Referee: Abstract, line 3. Did I miss something? Where do the authors discuss how the chemical composition of the plume changes downstream?

6 - Author's response: We thank the reviewer for this observation. However, the intention of this sentence appears to be different for this observation. What we want to address is a known fact, i.e., the Manaus pollution plume changes the atmospheric composition downstream.

7 - Comment from Referee: Abstract, lines 9-13. If the authors' model result indicates

that any breeze circulations might be confined below the 150-m level, was the design of the project, with flights limited to 500 m altitude, flawed? Also, was any effort done to consider lighter-wind vs. strong-wind conditions. One might think the breeze circulations would be more apparent in the former case. Please comment.

7 - Author's response: Please see comment/reply for 2 and 3 above. The study seeks to explain the observations, i.e., the study leads with the observations. The reviewer has a different kind of manuscript and scientific endpoints in mind (see comment/reply 2 and 3).

8 - Comment from Referee: Abstract lines 25-27. The authors conclude that "..most pollution was transported at heights well above the effects of the river breezes. . ." What then does this indicate about the wisdom of locating the vast DOE ARM resources at a single site at the surface (T3), downwind? Please comment on this and elaborate about what these sentences mean. How is the pollution 'information' communicated to the surface? Answering this question brings you straight back to trying to understand the differences between the CBL over land and whatever is present over the rivers.

8 - Author's response: The results suggest that observations at the T3 location are not greatly affected because the river breeze appears to push the pollution "up and over" rather than "capture and channel". The pollution is at the surface past the river and is sampled at T3 because of the atmospheric convection. Measurements at 500 m in the G-1 when location above T3 and measurements at the surface at T3 for the same time period are in agreement with respect to many atmospheric and particle tracers of pollution. The results are not shown in this manuscript because the topic is judged as out of scope (cf. Martin et al., BAMS, 2017).

9 - Comment from Referee: Line 48. Prof. Maria A. F. Silva Dias, has been cited for some time as "Silva Dias". Please follow this pattern, so that readers looking up cited papers will not get confused.

9 - Author's response: We thank the reviewer for the contribution. The alteration was

done in the manuscript.

10 - Comment from Referee: Line 166. On what time scale can one conclude that "carbon monoxide was mostly inert on the time scale of the simulations..."? Here are some competing scales: a) Mixing in the convective boundary layer $z_i/w^*$, where $w^*$, the convective velocity scale that depends on the surface buoyancy flux. This might be different over the river & over the land; b) the time it takes the air to advect from Manaus to the point of measurement; and of course, c) time of day (time since the surface layer became convective, though one might argue that over the river that layer was convective all night.

10 - Author's response: "Inert" refers to the chemistry, i.e., that the reaction of CO + OH —[O2]–> CO2 + HO2 has a much longer lifetime than the transport time within the high-resolution central box of the simulation. Chemical lifetimes of CO are typically 30 to 90 days on global scale in the lower atmosphere. The revised text changes: "carbon monoxide was mostly inert" to "carbon monoxide did not have significant chemical sinks".